

# Sex differences in strength at the shoulder: a systematic review

Tamar D. Kritzer[1], Cameron J. Lang[1], Michael W. R. Holmes[1] and Alan C. Cudlip[2]

[1] Department of Kinesiology, Brock University, St. Catharines, ON, Canada
[2] Department of Kinesiology and Health Sciences, University of Waterloo, Waterloo, ON, Canada

Corresponding author
Alan C. Cudlip,
accudlip@uwaterloo.ca

## ABSTRACT

**Background:** Understanding differential strength capability between sexes is critical in ergonomics and task design. Variations in study designs and outcome measures generates challenges in establishing workplace guidelines for strength requirements to minimize upper extremity risk for workers. The purpose of this systematic review was to collate and summarize sex differences in strength at the shoulder across movement directions and contraction types.

**Methods:** A total of 3,294 articles were screened from four databases (Embase, Medline, SCOPUS, and Web of Science). Eligibility criteria included observational studies, direct measurement of muscular joint, and healthy adult participants (18–65 years old). Strength outcome measures were normalized to percentages of male outputs to allow comparisons across articles.

**Results:** A total of 63 studies were included within the final review. Majority of articles observed increased strength in males; the gap between male–female strength was greater in flexion and internal/external rotation, with females generating ~30% of male strength; scaption strength ratios were most consistent of the movement groups, with females generating 55–62% of male strength.

**Conclusion:** Sex strength differences should be considered as an important factor for workplace task design as women are more at risk for occupational-related injuries than men in equivalent strength requirements. Differences in strength were not synonymous across motions; females demonstrated increased disparity relative to male strength in horizontal flexion/extension, forward flexion and internal/external rotation. Some movements had an extremely limited pool of available studies for examination which identified critical research gaps within the literature. Collating and quantifying strength differences is critical for effective workstation design with a range of users to mitigate potential overexertion risk and musculoskeletal injury.

## INTRODUCTION

Upper extremity use is an important part of our daily lives, but strength requirements for activities of daily living or work tasks may not be equal between sexes, representing differential capabilities. The shoulder joint is a complex and highly mobile joint that allows six degrees of movement. It is composed of four joints (glenohumeral, acromioclavicular, scapulothoracic, and sternoclavicular) that work in a rhythm to produce movements of the

shoulder including flexion/extension, abduction/adduction, scaption, horizontal flexion/extension (*i.e.*, horizontal adduction/abduction), and internal/external rotation. Various positions of the shoulder result in different strength and directional outcomes of the arm and shoulder. As well, certain positions lead to various outcomes such as rapidly fatiguing the musculature surrounding the shoulder. Biological differences between females and males play a major role in the risk factors of exposures to work-related musculoskeletal disorders (WMSD) (*Hooftman et al., 2009*). This WMSD risk varies depending on the body area affected; males are reported to be at higher risk for low back injuries while women report more neck-shoulder injuries (*Hooftman et al., 2009*). Multiple sources report that males consistently possess increased upper extremity strength compared to females (*Ivey, Calhoun & Rusche, 1985*; *Singh & Karpovich, 1968*; *Vianna, Oliveira & Araujo, 2007*). Strength differences are typically assessed using end-effector values such as force or torque. For the purpose of this article, strength is defined using end-effector outcomes such as force or torque, and is measured as either isometric (*i.e.*, static) or isokinetic (*i.e.*, dynamic) movements within a single plane (*i.e.*, flexion, adduction, *etc.*). Upper extremity strength tests have demonstrated a trend of lower strength capabilities of females during identical tasks compared to males, reporting 42–47% of male strength on average when completing various upper body strength tests (*Singh & Karpovich, 1968*). These differences may be based in physiology; males have larger muscle cross sectional areas (*Haizlip, Harrison & Leinwand, 2015*) and have ~75% more upper body muscle mass on average (*Lassek & Gaulin, 2009*). Anthropometric and muscle fibre characteristic differences between sexes justify significant differences of posture, strength, and fatigue resistance that may contribute to differences in functional capacity. This may represent a key factor demonstrating the differences of neck and shoulder WMSDs (*Côté, 2012*). Given the disparities in magnitude of male-female strength across upper extremity joints, a representative average is likely not specific enough for ergonomic consideration.

Understanding force capability is critical for digital modeling and effective workplace design. Biomechanists, ergonomists, engineers, and work task designers can leverage strength data information to ensure tasks incorporate capability requirements for both sexes and highlight directional differences at the shoulder. Digital human models (DHMs) are prevalent in physical ergonomics as a tool to proactively assess job demands and provide design considerations. To improve the fidelity of workplace simulations with DHMs, models need to provide valid posture and motion prediction (*Chaffin, 2005*). Many DHMs will calculate biomechanical loads from strength data and provide context relative to population specific strength or estimated tissue tolerances. The literature surrounding shoulder strength differences between males and females in the workplace is limited and has never been synthesized into a review.

As tools continue to improve and incorporate both sexes, a need exists in effectively quantifying these outputs; however, the range of tools and techniques used to date has made interpretation challenging. Muscular strength has been defined as a measurement of maximal voluntary isometric contraction (MVC) and the maximum force produced against an external resistance (*Schoenfeld et al., 2021*). Females have been demonstrated to produce less torque and power during a functional work task compared to males, even

when normalized to perceived work intensity (*Esmail, Bhambhani & Brintnell, 1995*). To date, there has been substantial variability in the methods used to quantify sex differences in strength, including end effector forces in Newtons (N) (*Collins & O'Sullivan, 2018*; *Coury et al., 1998*; *Hills & Bohannon, 1992*; *McKay et al., 2017*; *Yates et al., 1980*), torques in Newton-meters (Nm) (*Faber, Hansen & Christensen, 2006*; *Kim et al., 2009*; *Lannersten et al., 1993*; *MacDonell & Keir, 2005*; *Roy et al., 2009*), or alternative outcome measures, such as those normalized to participant body mass (*Marcondes et al., 2019*; *Riemann et al., 2010*). These differences between outcome measures when paired with methodological techniques presents a challenge to ergonomists, work task designers and ergonomic tools; development of a review to simplify these methodological differences and identify strength differences could prove beneficial to these parties. The purpose of this work was to critically assess the current literature and complete a systematic review to establish strength differences between sexes at the shoulder.

## METHODS

### Search strategy

A search was completed by considering the main topic of interest and selecting relevant keywords to efficiently extract articles from each database. Searches were conducted from four databases: Embase, Medline, SCOPUS, and Web of Science. The search consisted of three parts; identification of the shoulder as the relevant body region, combinations of terms commonly used to describe strength, and inclusion of both sexes. Year-bounds were not set for this search. The search strategy was critiqued and revised by the institutional Library staff to formulate a finalized search string (Table 1). Search strategies were completed with the assistance of the institutional Librarian services and all database searches were completed on June 13, 2022. The study was registered with PROSPERO (ID: CRD42022339026).

### Eligibility criteria

All extracted articles were screened for eligibility to be included in this review. Eligibility criteria included randomized control trials, controlled studies, cohort studies, pilot studies, and theses. Studies were considered if they included healthy participants with no previous upper extremity injury or deformity between the ages of 18–65 years old. This study focused on strength differences of adults and those within the "working population" range since the goal of the article is to utilize this information in future ergonomic task design. Only studies measuring muscular strength directly (*e.g.*, Newtons (N), Newton-meters (Nm), kilograms (kg), *etc.*) were deemed eligible. Exclusion criteria comprised of books, editorials, dissertations, or if the article was not published in English. Further studies were excluded if the following parameters were present: incorrect study design (*i.e.*, did not perform a shoulder-isolated isometric or isokinetic task within a single plane, or, results not split by sex), wrong outcomes (*i.e.*, did not directly measure muscular strength), and wrong patient population.

**Table 1 Search string entries for all databases.**

| | Embase: 1,980 | Medline: 1,007 | SCOPUS: 439 | Web of science: 1,131 |
|---|---|---|---|---|
| Concept 1: Upper extremity | Shoulder<br>Exp shoulder/ | Shoulder<br>Exp shoulder | (shoulder) | Shoulder |
| Concept 2: Strength | (Muscle or joint or isometric or isokinetic or isotonic or dynamic)<br>(Strength or force or moment or torque)<br>Exp Muscle Contraction/<br>Exp Muscle strength/<br>Exp Exercise/ | (Muscle or joint or isometric or isokinetic or isotonic or dynamic)<br>(Strength or force or moment or torque)<br>Exp Muscle Contraction/ or exp Muscle Strength/ or exp Physical Exertion | (Muscle AND strength OR muscle AND force OR muscle AND moment OR muscle AND torque OR joint AND strength OR joint AND force OR joint AND moment OR joint AND torque OR isometric AND strength OR isometric AND moment OR isometric AND torque OR isokinetic AND strength OR isokinetic AND force OR isokinetic AND moment OR isokinetic AND torque OR isotonic AND strength OR isotonic AND force OR isotonic AND moment OR isotonic AND torque OR dynamic AND strength OR dynamic AND force OR dynamic AND moment OR dynamic AND torque) | (Muscle or joint or isometric or isokinetic or isotonic or dynamic) AND (Strength or force or moment or torque) |
| Concept 3: Sex differences | (Woman or women or female*)<br>(Man or men or male*)<br>(sex or gender) | (Woman or women or female*)<br>(Man or men or male*)<br>(sex or gender) | (Woman or women or female*)<br>(Man or men or male*) | (Woman or women or female*) AND (Man or men or male*) OR (sex or gender) |

**Note:**
* indicates the use of a special character in search string

## Methodological approach

A multi-step screening process was utilized to derive the final selection of articles for full-text analysis and data extraction. All database searches were completed by a single researcher (AC) on the same day followed by a double-blind screening from two researchers (TK and CL). Articles that met the search term criteria were extracted from each database and collated into Covidence (Veritas Health Innovation, Melbourne, Australia) to manage the screening process. Following removal of duplicates, title/abstract screening was completed independently by the two reviewers according to the inclusion/exclusion criteria. Conflicts regarding inclusion/exclusion decisions were resolved with a meeting between the two reviewers and MH until all disagreements were settled. MH acted as the referee between all disagreements. Following initial screening, eligible articles underwent full-text screening by the same reviewers with identical inclusion/exclusion criteria and conflict resolution methods as the first round. All articles that passed the second round of screening were the final articles included for data extraction. A PRISMA flow diagram outlines the inclusion strategy (Fig. 1).

## Assessment of methodological quality of risk of bias

All included studies were assessed using the Risk of Bias In Non-Randomised Studies (ROBINS-I) (Table 2). The assessment consisted of seven risk domains: bias due to confounding, participation population, intervention classifications, intended intervention deviations, absence of outcome data, measurements of outcomes, and reported results
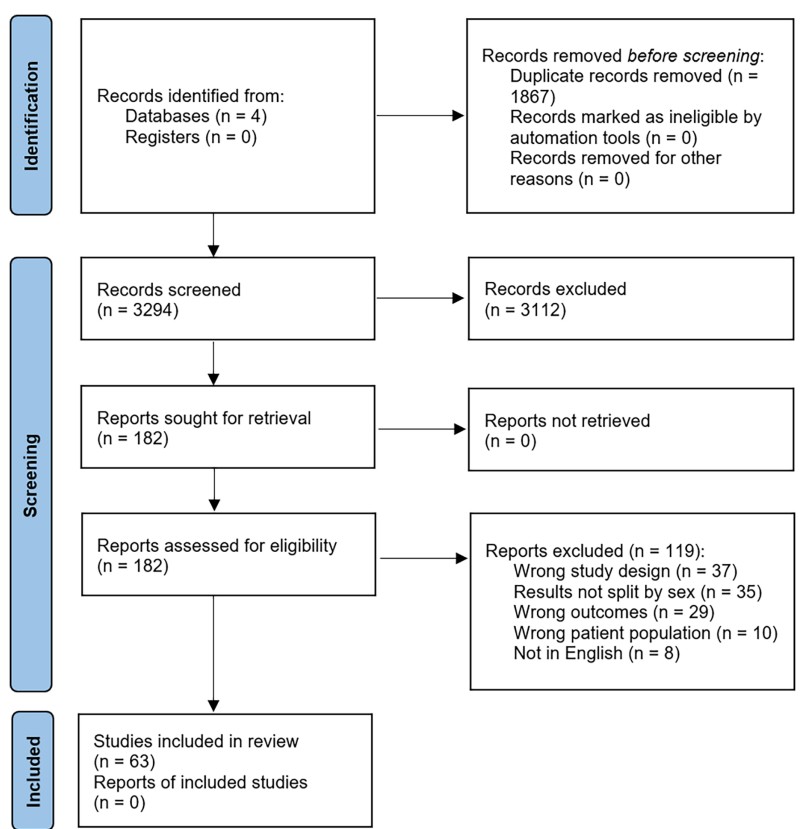

**Figure 1** **Preferred Reporting Items for Systematic Reviews and Meta-Analyses (PRISMA) flowchart describing the screening process and inclusion of eligible articles for this review.**

selections. Each study was assigned a level of bias per domain; escalating ratings were described as low (L), moderate (M), and serious (S). Bias ratings were collated, and article risk of bias was based on the highest rating within the seven domains.

## RESULTS

### Article selection

4,557 articles were extracted from the initial database retrieval for screening (Embase (1,980 articles), Medline (1,007 articles), SCOPUS (439 articles), and Web of Science (1,131 articles)). A total of 3,294 articles were screened with duplicates removed, where then 182 full-text articles were assessed for eligibility based on the details shown above. From this, a total of 63 articles were included and used for the assessment of the results. These articles exhibit experimental and observational studies that isolate movements of the shoulder to directly measure joint strength. A detailed flowchart of article screening can be found in Fig. 1.

All literature criteria for inclusion included peer-reviewed journal articles and conference writings/proceedings. A study was considered eligible if they contained a participant population without musculoskeletal injury or disease and contained both male and female participants within the working-age range (18–65 years). Articles were

**Table 2  Risk of bias assessments using ROBINS-I for included articles.**

| Title | Bias 1 | Bias 2 | Bias 3 | Bias 4 | Bias 5 | Bias 6 | Bias 7 | Overall score |
|---|---|---|---|---|---|---|---|---|
| Alizadehkhaiyat et al. (2014) | L | L | L | L | L | L | L | L |
| Andrews, Thomas & Bohannon (1996) | L | L | L | L | L | L | L | L |
| Aydin et al. (2001) | L | L | L | L | L | L | L | L |
| Bäckman et al. (1995) | L | L | L | L | L | L | L | L |
| Barnekow-Bergkvist et al. (2004) | L | L | L | L | L | L | L | L |
| Barrenetxea-Garcia et al. (2019) | L | L | L | L | L | L | L | L |
| Buśko & Gajewski (2011) | L | L | L | L | L | L | L | L |
| Cahalan, Johnson & Chao (1991) | L | L | L | L | L | L | L | L |
| Chezar et al. (2013) | L | L | L | L | L | L | L | L |
| Collins & O'Sullivan (2018) | L | L | L | L | L | L | L | L |
| Cools et al. (2016) | L | L | L | L | L | L | L | L |
| Danneskiold-Samsøe et al. (2009) | L | M | L | L | L | L | L | M |
| Douma et al. (2014) | L | L | L | L | L | L | L | L |
| Ellenbecker & Roetert (2003) | L | L | L | L | L | L | L | L |
| Eren et al. (2019) | L | L | L | L | L | L | L | L |
| Faber, Hansen & Christensen (2006) | L | M | L | L | L | L | L | M |
| Ferreira, Górski & Gajewski (2020) | L | L | L | L | L | L | L | L |
| Coury et al. (1998) | L | L | L | L | L | L | L | L |
| Guirelli et al. (2021) | L | L | L | L | L | L | L | L |
| Hageman et al. (1989) | L | L | L | L | L | L | L | L |
| Harbin, Leyh & Harbin (2020) | L | M | L | L | L | L | L | M |
| Harbo, Brincks & Andersen (2012) | L | L | L | L | L | L | L | L |
| Hartsell (1998) | L | L | L | L | L | L | L | L |
| Hill, Pramanik & McGregor (2005) | L | L | L | L | L | L | L | L |
| Hills & Bohannon (1992) | L | L | L | L | L | L | L | L |
| Holzbaur et al. (2007) | L | L | L | L | L | L | L | L |
| Huberman, Scales & Vallabhajosula (2020) | L | M | L | L | L | L | L | M |
| Hughes et al. (1999) | L | L | L | L | L | L | L | L |
| Ivey, Calhoun & Rusche (1985) | L | L | L | L | L | L | L | L |
| Khalaf & Parnianpour (2001) | L | L | L | L | L | L | L | L |
| Kim et al. (2009) | L | L | L | L | L | L | L | L |
| Kramer & Ng (1995) | L | L | L | L | L | L | L | L |
| Kramer & Ng (1996) | L | L | L | L | L | L | L | L |
| Koski & McGill (1994) | L | L | L | L | L | L | L | L |
| Lannersten et al. (1993) | L | L | L | L | L | L | L | L |
| Lindström et al. (2003) | L | L | L | L | L | L | L | L |
| Lorenzo & Nunez (2021) | L | L | L | L | L | L | L | L |
| MacDonell & Keir (2005) | L | L | L | L | L | L | L | L |
| Maddux, Kibler & Uhl (1989) | L | L | L | L | L | L | L | L |
| Magnusson et al. (1995) | L | M | L | L | L | L | L | M |
| Marcondes et al. (2019) | L | L | L | L | L | L | L | L |

| Table 2 (continued) | | | | | | | | |
|---|---|---|---|---|---|---|---|---|
| Title | Bias 1 | Bias 2 | Bias 3 | Bias 4 | Bias 5 | Bias 6 | Bias 7 | Overall score |
| *Mayer et al. (1994)* | L | L | L | L | L | L | L | L |
| *McKay et al. (2017)* | L | L | L | L | L | L | L | L |
| *McMaster, Long & Caiozzo (1992)* | L | L | L | L | L | L | L | L |
| *Meldrum et al. (2007)* | L | L | L | L | L | L | L | L |
| *Motta et al. (2019)* | L | L | L | L | L | L | L | L |
| *Murgia et al. (2018)* | L | L | L | L | L | L | L | L |
| *Murray et al. (1985)* | L | L | L | L | L | L | L | L |
| *Nyberg et al. (2014)* | L | L | L | L | L | L | L | L |
| *Pontillo & Sennett (2020)* | L | M | L | L | L | L | L | M |
| *Reid et al. (1989)* | L | L | L | L | L | L | L | L |
| *Riemann et al. (2010)* | L | L | L | L | L | L | L | L |
| *Roy et al. (2009)* | L | L | L | L | L | L | L | L |
| *Carrascosa-Sanchez et al. (1999)* | L | L | L | L | L | L | L | L |
| *Carrascosa-Sanchez et al. (2023)* | L | L | L | L | L | L | L | L |
| *Shklar & Dvir (1995)* | L | L | L | L | L | L | L | L |
| *Smith et al. (2001)* | L | L | L | L | L | L | L | L |
| *Stausholm et al. (2021)* | L | L | L | L | L | L | L | L |
| *van Cingel et al. (2007)* | L | L | L | L | L | L | L | L |
| *Van Harlinger, Blalock & Merritt (2015)* | L | L | L | L | L | L | L | L |
| *van Meeteren, Roebroeck & Stam (2002)* | L | L | L | L | L | L | L | L |
| *Westrick et al. (2013)* | L | L | L | L | L | L | L | L |
| *Yates et al. (1980)* | L | L | L | L | L | L | L | L |

**Note:**
L , Low risk; M, Moderate risk; H, High risk; article scores were represented as the highest score received across domains.

excluded if the participant pool had a chronic disease or musculoskeletal injury to the torso or upper extremity, contained participants from a single-sex, were a systematic review, or were not written in English. Participant pools using specific populations (*e.g.* an athletic group) have been identified and are outlined within the supplementary appendices, where extracted variables from each article are included in depth. The current work did not include multi-joint movements such as pushing or pressing motions as they may have been limited by strength outside of the shoulder complex, but represents a potentially valuable insight into human capacity. Detailed results of the variables extracted from the included studies can be found in the supplementary reading (Appendix Table S1).

**Body region/movements**

Articles included in this analysis were subdivided into nine sections based on planar movements: flexion ($n = 25$), extension ($n = 14$), scaption ($n = 2$), abduction ($n = 34$), adduction ($n = 18$), horizontal flexion/extension ($n = 2$), internal rotation ($n = 36$), and external rotation ($n = 37$). In 35 articles examined isometric movements, 20 articles examined isokinetic movements, and nine articles examined both types of movements.

Shoulder flexion/extension is known as raising/lowering a straight arm up in the sagittal plane, abduction/adduction is raising/lowering a straight arm up the frontal plane of the body, and scaption is raising a straight arm up in a diagonal position ~40° anterior of adduction. Horizontal flexion/extension is known as moving a straight arm medial to lateral to medial while in front of the body. Lastly, internal/external rotation produces a rotating movement of a straight arm, leading to the supination and pronation of the hand. All movements can be seen in Fig. 2. Expanded details can be found in the supplementary reading (Appendix Table S1).

## Measurement methodology

Multiple measurement tools were used to assess shoulder strength throughout the studies (Appendix Table S1). A total of 38 studies used an isokinetic or isometric dynamometer while 14 studies used a hand-held dynamometer. A small number of studies used alternative measurement tools, including gauge dynamometers (2), an electromechanical force transducer (1), an electromagnetic dynamometer (1), a load cell (1), a torque meter (2), a myometer (1), a quantitative muscle assessment system (1), a manual muscle tester (1), and a custom-made resting equipment (1).

Multiple strength outcome measures were observed across articles (Appendix Table S1). The most common measurement units used were Newtons (N), Newton-meters (Nm), and kilograms (kg) when measuring shoulder isometric and isokinetic strength. Two alternate measurement units described in the literature included strength normalized to percentage of body mass, and foot-pounds of torque (Appendix Table S1). The majority of the movement types used similar measurement units across all studies. Flexion and extension, adduction and abduction, and internal and external shoulder rotation all used N, kilogram per centimeter (kg-cm), Nm, foot-pounds (ft-lbs), and percent of body mass. Other movements such as scaption and horizontal flexion and extension used more consistent measurement units, using only Newtons, Newton-meters per kilogram, and pounds. Due to the variation of outcome measurements, strength data was normalized to a male: female percentage for comparison. Within each study, female results were divided by male results, then multiplied by 100 to conclude a percentage of strength for comparison.

## Directional outcomes

Most movement directions demonstrated similar outcome ranges. Each movement direction's outcome range based on the articles included in this systematic review can be found in Fig. 3. Outcomes described in Fig. 3 represent all shoulder joint movements demonstrated within the literature. The numbers shown in the chart are representative of the number of studies that examined each of the movements shown. For example, 37 articles had participants perform movements using external rotation of the shoulder. The lines represent the range of female strength that was measured in each of the articles that were examined. When examining abduction of the shoulder across all 34 articles, female strength was represented as 30–130% of male strength, with different tasks for each of these movement patterns.

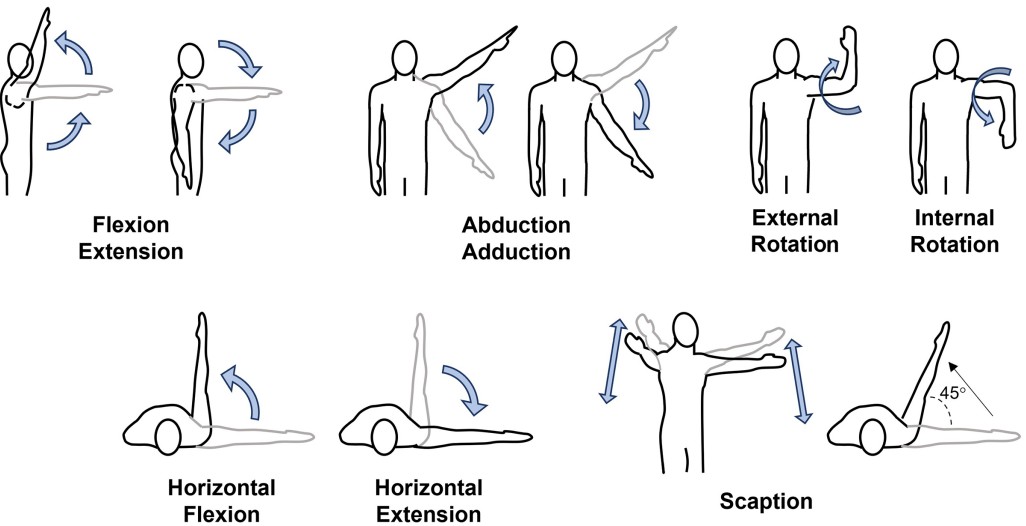

**Figure 2 Representative motions for the nine shoulder planar movements included in the review.**

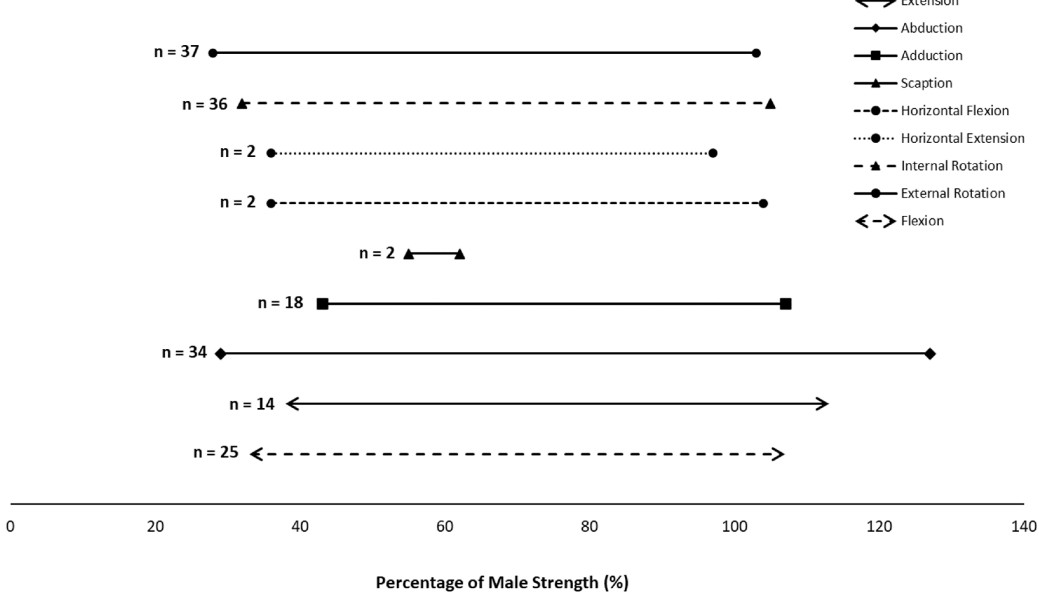

**Figure 3 Representative outcome ranges for shoulder articulations included in this systematic review.**

### Adduction and abduction

Generally, males generated greater adduction and abduction strength, with females typically producing ~40–80% of male abduction strength and ~45–65% of adduction strength. Expanded details can be found in the supplementary reading (Appendix Tables S2 and S3). Obtained female strengths ranged from 29–127% (*Cahalan, Johnson & Chao, 1991*; *Magnusson et al., 1995*) and 43–107% for abduction (Fig. 4A) and adduction

a)

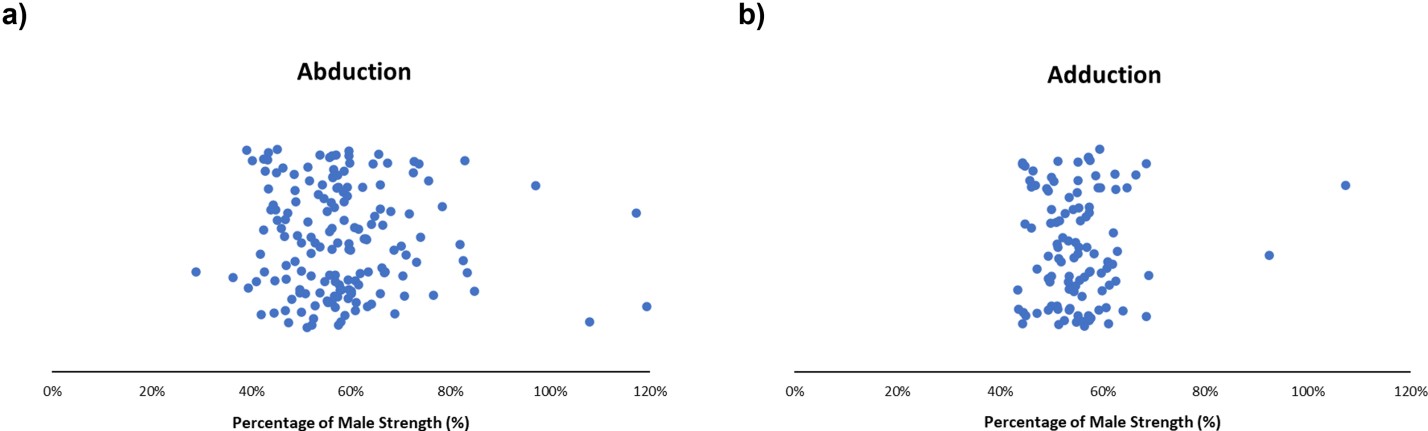

b)

**Figure 4 Sex strength differences of abduction (left) and adduction (right).** Female strength metrics ranging from (A) 29–127% and (B) 43–107% of male strength, respectively. Articles in this review demonstrate that female strength centralized at ~40–80% (abduction) and ~45–65% (adduction) of male strength.

(Fig. 4B), respectively (*Cahalan, Johnson & Chao, 1991*; *Huberman, Scales & Vallabhajosula, 2020*). For abduction and adduction during isokinetic motions, females produced 40–70% of male strength when working at 30, 60, 120 and 180 degrees/second (*Carrascosa-Sanchez et al., 1999*; *McMaster, Long & Caiozzo, 1992*; *Shklar & Dvir, 1995*). Three articles had female abduction strength exceed male abduction strength (*Huberman, Scales & Vallabhajosula, 2020*; *Magnusson et al., 1995*; *Murgia et al., 2018*). Two instances occurred during isometric abduction: with generated female strengths of 108% of males (*Huberman, Scales & Vallabhajosula, 2020*), and females resulting in 127% of male strength (Appendix Table S2) (*Magnusson et al., 1995*). One instance occurred in an older population during isokinetic abduction at 60°/s with females generating 119% of male strength (Appendix Table S2) (*Murgia et al., 2018*). In isometric adduction contractions, females produced 45–60% of male strength (*Holzbaur et al., 2007*; *Meldrum et al., 2007*; *Murray et al., 1985*). A single study observed female adduction strength exceeding male adduction strength, with females producing 107% of males (48.20 ± 16.15 lbs for females, 44.93 ± 16.09 lbs for males) (*Huberman, Scales & Vallabhajosula, 2020*) (Appendix Table S3).

### Scaption

Across conditions, females generated 55–62% of male scaption strength (Fig. 5) (*Alizadehkhaiyat et al., 2014*; *Eren et al., 2019*). Expanded details can be found in the supplementary reading (Appendix Table S4). Sex differences were smaller in the empty can test (88.92 ± 23.96 N *vs.* 54.95 ± 14.18 N; 62% of male strength) than the palm down test (98.89 ± 25.99 N *vs.* 54.26 ± 10.8 N; 55%) (*Eren et al., 2019*) (Appendix Table S4). Other work observed females generating 61% of male scaption strength (*Alizadehkhaiyat et al., 2014*).

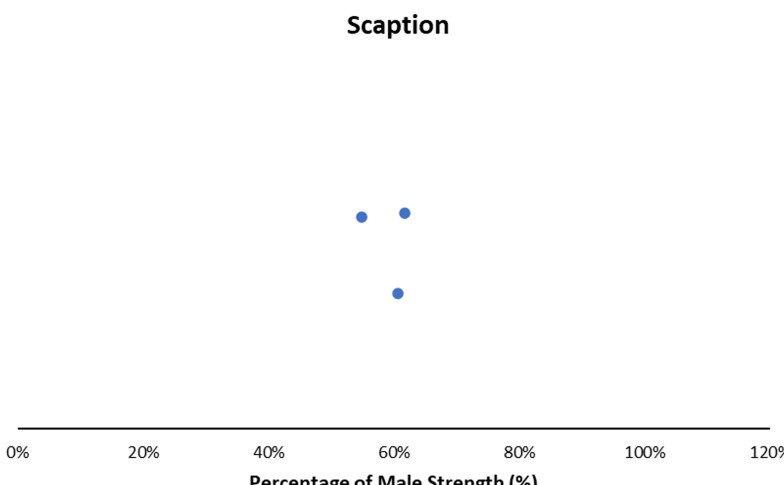

**Figure 5 Sex strength differences of scaption.** Female strength metrics ranging from 55–62% of male strength.

### Flexion and extension

Males demonstrated greater shoulder torque than females in flexion and extension across most articles. Females typically produced ~40–70% of flexion strength (Fig. 6A) and ~40–60% of extension strength (Fig. 6B) of males; obtained female strengths ranged from 33–107% (*Huberman, Scales & Vallabhajosula, 2020*; *Khalaf & Parnianpour, 2001*) and 38–113% for flexion and extension, respectively (*Hughes et al., 1999*; *Khalaf & Parnianpour, 2001*). Expanded details can be found in the supplementary reading (Appendix Tables S5 and S6). A single isometric study demonstrated females exceeding male flexion (Appendix Table S5) and extension (Appendix Table S6) strength; flexion strengths were 39.39 ± 10.85 and 36.93 ± 10.82 lbs and 54.41 ± 14.57 *vs*. 53.25 ± 14.52 (Nm) in extension for males and females, respectively (*Huberman, Scales & Vallabhajosula, 2020*). Isokinetic flexion articles tended to report greater disparities in reported strength compared to isometric articles. Isometric articles demonstrated male and female flexion strength of 22.4 ± 4.7 kg and 9.8 ± 3.5 kg respectively (*Van Harlinger, Blalock & Merritt, 2015*) compared to isokinetic flexion articles which demonstrated male and female flexion strength of 102.3 ± 7.4 and 72.4 ± 8.2 % body mass at 60°/s (*Marcondes et al., 2019*).

### Horizontal flexion and extension

Males generated increased horizontal flexion/extension strength in most included articles. On average, females produced roughly 40–55% of horizontal flexion strength (Fig. 7A) and 40–50% of horizontal extension of males (Fig. 7B). Expanded details can be found in the supplementary reading (Appendix Tables S7 and S8). Obtained female strengths ranged from 36–104% (*Huberman, Scales & Vallabhajosula, 2020*; *Van Harlinger, Blalock & Merritt, 2015*) and 36–97% (*Huberman, Scales & Vallabhajosula, 2020*; *Van Harlinger, Blalock & Merritt, 2015*) for horizontal flexion and extension respectively. A single isometric study observed elevated female horizontal flexion strength relative to other

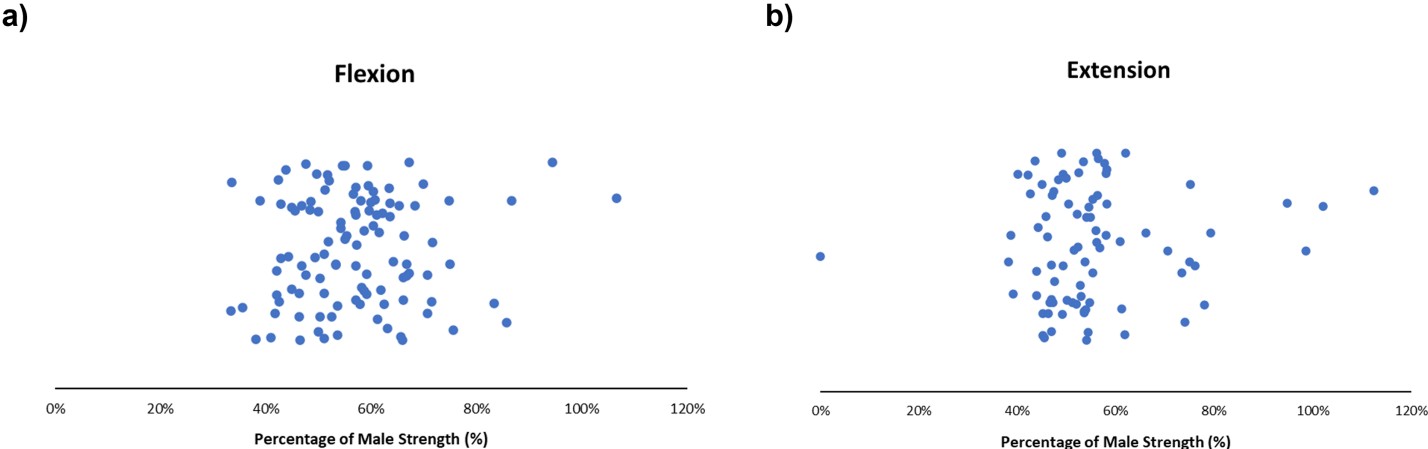

**Figure 6 Sex strength differences of flexion (left) and extension (right).** Female strength metrics ranging from (A) 33–107% and (B) 38–113% of male strength, respectively. Articles in this review demonstrate that female strength centralized at ~40–70% (flexion) and ~40–60% (extension) of male strength.

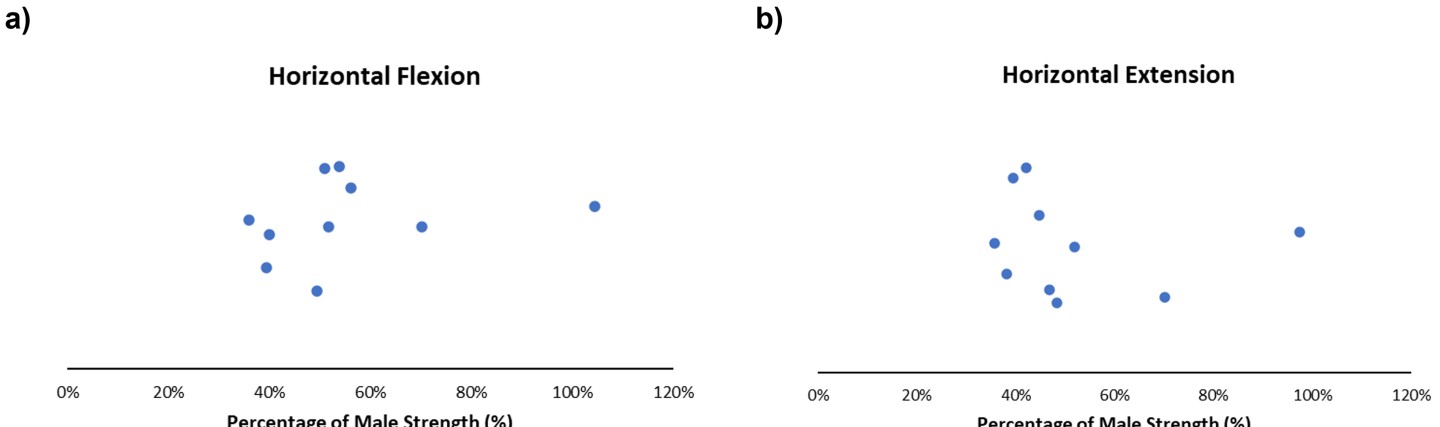

**Figure 7 Sex strength differences of horizontal flexion (left) and horizontal extension (right).** Female strength metrics ranging from (A) 36–104% and (B) 36–97% of male strength, respectively. Articles in this review demonstrate that female strength centralized at ~40–55% (horizontal flexion) and ~40–50% (horizontal extension) of male strength.

articles, with females generating 97% of male horizontal flexion strength, when normalized (female = 43.59 ± 14.95 lbs; males = 41.80 ± 14.90 lbs) (*Huberman, Scales & Vallabhajosula, 2020*). Only isometric force data was presented for horizontal flexion and extension (Appendix Tables S7 and S8).

### Internal/external rotation

On average, females typically produced ~40–70% of male internal rotation strength (Fig. 8A) and ~45–80% of external rotation strength (Fig. 8B). Expanded details can be found in the supplementary reading (Appendix Tables S9 and S10). Obtained female strengths ranged from 32–105% (*Huberman, Scales & Vallabhajosula, 2020*; *Van Harlinger, Blalock & Merritt, 2015*) and 28–103% (*Cahalan, Johnson & Chao, 1991*; *Huberman, Scales & Vallabhajosula, 2020*) for internal rotation and external rotation, respectively. During isometric movements, females consistently demonstrated
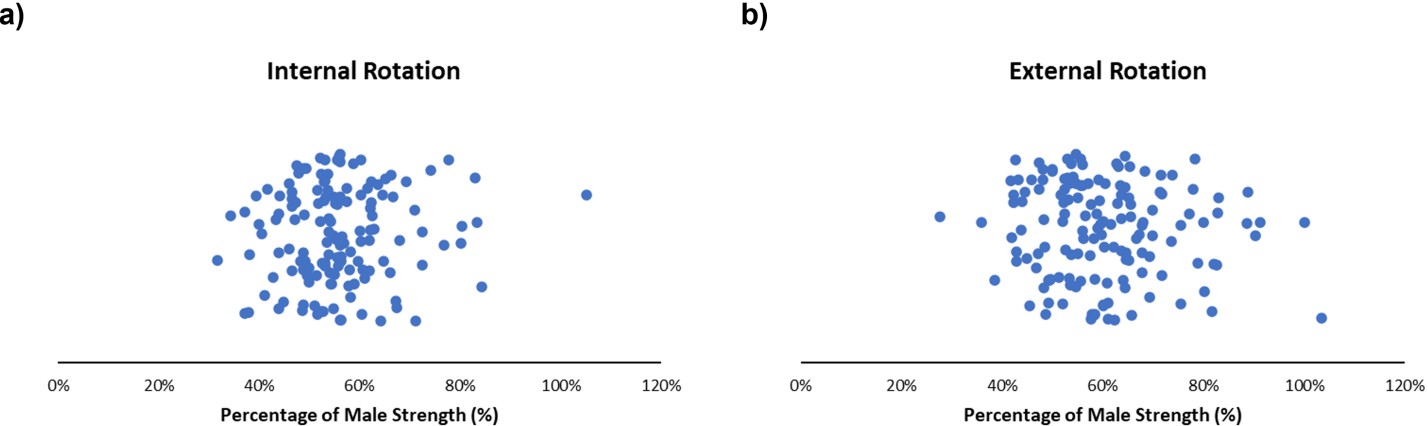

**Figure 8 Sex strength differences of internal rotation (left) and external rotation (right).** Female strength metrics ranging from (A) 32–105% and (B) 28–103% of male strength, respectively. Articles in this review demonstrate that female strength centralized at ~40–70% (internal rotation) and ~45–80% (external rotation) of male strength.

approximately half the strength outputs compared to males (*Lannersten et al., 1993*; *Roy et al., 2009*). Males generated external rotation torques of 26.4 ± 8.6 Nm compared to 14.9 ± 3.4 Nm from females (*Lannersten et al., 1993*). During internal rotation and external rotation, females generated 53% and 57% of male strength (Appendix Tables S9 and S10) (*Roy et al., 2009*). Isokinetic testing used in included studies ranged from 60–180 degrees/s; as speeds increased, strength outputs decreased, with similar decreases between sexes (*Cahalan, Johnson & Chao, 1991*; *Marcondes et al., 2019*; *Shklar & Dvir, 1995*).

## Bias and quality of assessments

The methodological quality of each study was represented by a score of low (L), moderate (M), or serious risk of bias (S), with an overall bias rating given for each article. The methodological quality scores and rating of bias is located in Table 2. Risk of bias assessments of each article conducted using the ROBINS-I identified six articles having a moderate risk of bias in the domain of selection of participants (*Danneskiold-Samsøe et al., 2009*; *Ellenbecker & Roetert, 2003*; *Faber, Hansen & Christensen, 2006*; *Harbin, Leyh & Harbin, 2020*; *Huberman, Scales & Vallabhajosula, 2020*; *Magnusson et al., 1995*; *Pontillo & Sennett, 2020*).

## DISCUSSION

Anatomical and biomechanical sex differences exist within the shoulder that identify strength inequalities between males and females. The primary objective of this systematic review was to examine strength differences between sexes at the shoulder for various movement direction and contraction types. The heterogeneity in methodologies and outcome measures across articles prevented ease of comparison or traditional statistical analysis and may have been a factor in the substantial ranges of relative strengths when normalized to percentages of males. This review provides strength ranges across all articles, typically demonstrating that on average females generate lower strength in flexion (57%), extension (56%), abduction (59%), adduction (55%), scaption (59%), horizontal flexion

(55%), horizontal extension (51%), and internal rotation (56%), and external rotation (60%) compared to male strength percentages. Specific articles encompassing each directional movement type can be seen in more detail in the Appendix Tables S3–S10. The majority of the included articles concluded that males generated increased strength compared to females across most joint movements examined regardless of age, weight, height, and athletic experience. Ergonomist and workplace designers should acknowledge the decrease in relative strength based on the movement direction and contraction type to improve occupational workplace guidelines. Although with articles typically focused on flexion/extension, abduction/adduction, and internal/external rotation, workplace and task design should account for the lack of data within other movement directions and contraction types when considering the decrease in relative strength between sexes.

Disparities in strength between sexes were slightly larger in flexion, internal rotation and external rotation. Across movements, obtained female strengths ranged from 28% to 127% of male strength with some strength equivalencies. Abduction, flexion, extension, horizontal flexion, horizontal extension, internal rotation, and external rotation presented similar relative female strength of ~30 to >100%. Improved female relative strength was observed in adduction and scaption, with outcome measured ranging from ~45% to >100%. Scaption presented the smallest range for female strength, where females generated 55–62% of male strength; abduction presented the largest range of 29–127% of male strength across all articles (Fig. 3). Few exceptions demonstrated an increased female strength compared to males across movements; each of these studies included potential bias in the results. One article measuring isometric strength presented participant selection bias when recruiting, as participants had to be graduate level competitive swimmers from a single New York metropolitan area (*Magnusson et al., 1995*). Another article measuring isokinetic strength only presented greater female strength results in their older adult participant sample, suggesting possible anthropometric changes with aging (*Murgia et al., 2018*). A single article demonstrated greater female strength across all joint movements (*Huberman, Scales & Vallabhajosula, 2020*), but participants were recruited from circus studios across the United States with 157 females and 30 males recruited; differences in participant pool from a unique population may have affected their results. Other than these few exceptions, majority of the results demonstrated greater strength in males compared to females across movements.

Anatomical and biomechanical studies identify increased reporting of neck-shoulder symptoms in females. With the same job responsibilities, women have a higher risk for occupational related injuries due discrepancies between the workplace and their anthropometric and physiological differences (*Treaster & Burr, 2004*). Most workplaces are designed based on male anthropometry, creating disadvantages for smaller individuals, who are typically female (*Treaster & Burr, 2004*). These ill-fitting stations tend to generate various movement compensations, leading to increases in postural strain (*Treaster & Burr, 2004*). Increased laxity of the female shoulder compared to males can leads to an increased sensitivity to tendon overstretching resulting in an increased risk of instability and injury (*Maier et al., 2022*). Task demand and design should consider force production differences at the various task postures. For the same task, women must use increased muscle activity,

causing them to work at efforts closer to their maximal capacity (*Maier et al., 2022*). These factors along with the strength differences of each movement direction should be considered as differences in work-related functional capacity imply the higher risk of shoulder injury development (*Maier et al., 2022*).

Included articles predominantly focused on a small number of isometric and isokinetic movements, highlighting research scarcity in some areas. The majority of articles included were focused on ab/adduction, flexion/extension, and internal/external rotation, suggesting large research gaps in scaption, horizontal flexion, and horizontal extension movements. The sparsity of articles for these movements may cause ambiguity towards overarching results and should be considered with caution. Horizontal flexion and horizontal extension movements presented lower ranges compared to the other movements with most results presenting as ~36–50% female-male strength (*Huberman, Scales & Vallabhajosula, 2020*; *Van Harlinger, Blalock & Merritt, 2015*), but all values were extracted from a single article and should be interpreted with caution. Articles presenting isometric results demonstrated higher relative female strength compared to isokinetic results of abduction, adduction, flexion, extension, internal rotation, and external rotation. Similarities of isokinetic speed selections were limited. The majority of the isokinetic movement articles examined strength at 60°/s and 180°/s; the 12 other isokinetic movement speeds had a limited amount (<3) of articles that used the same methodology, preventing direct comparisons. It is apparent that sex-based analyses are necessary to improve the understanding of work-related strength differences and how it affects exposures to musculoskeletal disorders.

Understanding sex differences in upper extremity capability is a critical aspect of the advancement of workplace design, and allows ergonomists and engineers to create workspaces that are ideal for all individuals and advances ergonomic guidelines and standards. When we examine the strengths demonstrated by both males and females during various shoulder movements, females demonstrated approximately of 50% of male strength across multiple articles during elevation/depression and flexion/extension movements (*Shklar & Dvir, 1995*; *Reid et al., 1989*; *van Meeteren, Roebroeck & Stam, 2002*). This becomes problematic for female employees, especially during lifting tasks, as there will be an increased risk of injury or the inability to properly perform tasks. If the task involves pushing or pulling a cart with a load of 50 kg or more, females reported pain or injury six times more frequently than males (*Hoozemans et al., 2002*). It was also shown that females produce less torque and power during a functional work task compared to males, even when normalized to perceived work intensity (*Esmail, Bhambhani & Brintnell, 1995*). When both males and females were working in a rubber plant factory and performing the same task, posture and movement stayed relatively similar, however, females had greater shoulder muscle activation (*Nordander et al., 2008*). The information demonstrated in this study will assist in updating ergonomic guidelines to better understand sex differences and incorporate them into best practices design.

## LIMITATIONS

There are limitations to be considered for this review. A high variability of the methodology of strength measures limited comparisons of results. Postural differences and isokinetic speed selections may confound interpretation of these findings. The lack of normalization to body weight within research articles increased heterogeneity of results preventing comparison of results; all article results were scaled to a percentage of male strength to normalize force metrics for the systematic review. Disparities in the number of articles included for each force direction may have affected outcomes and demonstrates a strong need for additional research. Future articles should consider normalizing results to body weight for proper comparison of sexes or replicating past strength comparison articles to reduce the force metrics introduced and increase the quantity of articles focusing on movements lacking research. Additionally, this work does not consider demographic differences such as race, employment, socioeconomic status, *etc*. Articles including injured populations were not examined within this review. Limb dominance was not considered as a factor related to upper limb strength within this review, future articles should consider investigating this factor. When available, both dominant and non-dominant information have been identified and included in the supplementary tables (Appendix Tables S2–S10). Outcome strength differences of injured populations may increase or decrease with history of injury. There is a need to continue to examine strength differences of various demographics and movement types.

## CONCLUSION

The purpose of this systematic review was to critically examine the strength differences between sexes for the shoulder joint of the upper extremity. Males have shown to generate more strength compared to females across most joint movements examined regardless of age, weight, height, and athletic experience. Although some studies identified greater female strength compared to males across movements, each included potential sources of bias which may have influenced the results. For this review, all results were normalized to a percentage of male strength due to the lack of homogeneity of research metrics. Relative female strength was generally higher in adduction and scaption; strength ratios did not appear to alter substantially in other directions. Female relative strength was generally higher in adduction and scaption; strength ratios did not appear to alter substantially in other directions. Discrepancies in force directions, movement types, and movement speeds may have affected outcomes demonstrating a strong need for additional research. This systematic review provides biomechanists, ergonomists, and work task engineers collated data that can be leveraged to design safe workspaces and tasks, as well as highlighting locations of scarcity within the research field. Future ergonomic task and/or workplace design should consider these directional effects in design.

## ACKNOWLEDGEMENTS

The authors would like to thank Ian D. Gordon and the Brock University Library for their expertise with evidence synthesis, database search term development, knowledge translation, and support for this research review.

### Funding

This research was funded by an NSERC Discovery Grant held by Michael W.R. Holmes (RGPIN 2023-04488). The funders had no role in study design, data collection and analysis, decision to publish, or preparation of the manuscript.

### Grant Disclosures

The following grant information was disclosed by the authors:
NSERC Discovery Grant: RGPIN 2023-04488.

### Competing Interests

Michael W. R. Holmes is an Academic Editor for PeerJ.

### Author Contributions

- Tamar D. Kritzer performed the experiments, analyzed the data, prepared figures and/or tables, authored or reviewed drafts of the article, and approved the final draft.
- Cameron J. Lang performed the experiments, analyzed the data, prepared figures and/or tables, and approved the final draft.
- Michael W. R. Holmes conceived and designed the experiments, authored or reviewed drafts of the article, and approved the final draft.
- Alan C. Cudlip conceived and designed the experiments, performed the experiments, analyzed the data, authored or reviewed drafts of the article, and approved the final draft.

### Data Availability

   The raw outcome data are available in the Supplemental Tables.

### Supplemental Information

Supplemental information for this article can be found online at http://dx.doi.org/10.7717/peerj.16968#supplemental-information.

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
