# Peer review of "Sex differences in strength at the shoulder: a systematic review"

_PeerJ, doi:10.7717/peerj.16968_

## Round 0.1 · original submission · Major Revisions

The reviewers generally supported publication of the manuscript, but requested more details on a number of points. Please address these points.

Reviewer 1 ·

Basic reporting

Minor comment on Figure 1: Some of the text is outside of the boxes/shapes.

Experimental design

This manuscript addresses an important and timely question. It is overall well written, and the synthesis of the results would add in interesting contribution to the area of sex-based analysis. However, there are some aspects of the review design that need further description and refinement in order to better understand the results and interpretation:

1. Strength is one of the primary concepts of this review, but I am not clear on how the authors are defining strength. Based on the results, the focus appears to be isometric and isokinetic strength; was that the purpose? Why not dynamic strength, i.e. differences in bench press 1RM or maximal lift capacity between sexes? It appears ‘dynamics’ was included in your search, but not addressed in the results. I suggest clearly defining your approach to “strength” to help the readers better understand the purpose of this review.

2. The inclusion and exclusion criteria are very vague. It is difficult to understand how you determined which studies made it to full text extraction. For example, what does “incorrect study design” mean? Which study designs were unacceptable? Also, “wrong outcomes” does not help us to understand what outcomes you were intending to examine (this may be tied to the definition of strength). These criteria should be clearly stated. You could consider a PICO table (Participant, Intervention, Comparison, Outcome), or even include a table/list of your inclusion/exclusion questions.

I have some other smaller comments outlined by section below.

Validity of the findings

No comment.

Additional comments

Methods
1. Search strategy – did you have any year bounds for your search?
2. Eligibility criteria, line 107 – please justify the 18-65 age bounds.

Results
1. Line 152 – “35 articles examined isometric movements, 20 articles examined isokinetic movements, and 9 articles examined both types of movements.” – again, why were no dynamic strength measures included?

Discussion
1. Line 288 – I am not sure it is accurate to say this review’s objective was to “identify” strength differences – perhaps Examine? Aggregate?
2. Lines 311-321 – this section adds great context to the review. However, to be clear, in all other articles except the 3 mentioned here, was female strength <100%? If so, it would be worthwhile plainly stating that.
3. Line 317 – “elderly participant sample” – your age bound was 65, so what are you considering elderly in the context of this review?
4. Lines 328-329 – I’m not sure what this statement adds or how it is relevant to the paragraph
5. Lines 336-339 – why were only these movements examined? Are there are no isometric/isokinetic/dynamic push/pull or lift/lower sex comparisons?
6. I would suggest adding a paragraph putting these results into context of other literature comparing sexes, such as sports performance or functional capacity normative data. It is not unexpected that women had lower strength outcomes than men, but was this level of difference expected or typical for other types of sex-based performance comparisons?

·

Basic reporting

This study comprehensively investigates sex differences in shoulder strength across various movement directions and contraction types. The systematic review collates data from a substantial number of papers and effectively normalizes strength outcome measures to the proportion of male outputs for meaningful comparisons. While the study carefully describes the research objectives and discussions, it is essential to provide detailed descriptions of certain processes to ensure accuracy and transparency in the review.

Abstract: Lines 22-23: Systematic reviews aim to collect evidence that meets pre-specified eligibility criteria and, unlike meta-analyses, will not be quantified. I recommend changing to "summary.”

Abstract: Lines 24-26: The Methods section should specify the information sources (e.g., databases, registers), inclusion and exclusion criteria for the review, and the methods used to present the results.

Abstract: Lines 24-26: I would suggest moving the total number of studies included, to the Results section.

Experimental design

Methods: This study was presented in accordance with PRISMA 2020. It is recommended that a complete search strategy for all databases be presented, including filters and limitations used.

Methods: I recommend providing detailed information in the Methods section regarding the data extraction process from the included studies. Specifically, it would be beneficial to outline the variables planned for extraction and specify the time points (e.g., baseline or post-intervention) for data extraction, particularly if randomized controlled trials or cohort studies were included.

Validity of the findings

Results: Because this systematic review covers a variety of study designs (randomized controlled trials, controlled studies, cohort studies, and so on), I suggest adding information on study designs that met the inclusion criteria. Clarifying this would help ensure the accuracy and transparency of the systematic review.

Discussion: Lines 291-295: I recommend adding a text that specifies the source of data for the mentioned percentages related to different directions of motion.

Discussion: Lines 313-321: The Results section did not include specific descriptions of participant characteristics (e.g., general population, participants with sports experience). Describing the characteristics of the participant population would be helpful in interpreting the discussion (e.g., the heterogeneity of the participants) of this manuscript.

Reviewer 3 ·

Basic reporting

The current systematic review aimed to investigate sex-based differences in shoulder strength across various movement directions and contraction types, with a focus on their implications for ergonomics and task design in the workplace. The study screened 3,294 articles from four databases, ultimately including 63 articles for the review. The strength measurements were normalized as percentages of male strength to facilitate comparisons between studies. The findings indicated that, in most cases, males exhibited greater shoulder strength compared to females. Please find the suggestions below:

Introduction
1. The shoulder joint is a highly mobile and complex joint that relies on a network of muscles to produce various movements. Adding the primary muscles responsible for different motions of the shoulder joint would help the readers and provide valuable information for further application of the current knowledge.
2. Regarding the horizontal flexion or extension, adding another term for this movement (horizontal adduction or abduction) is needed.
3. Since the shoulder joint is complex, adding more explanation about the anatomical structure of the shoulder complex is important. For example, the current review has also reported the scaption, a motion related to both the shoulder joint and scapular (shoulder blade) movement. Scaption is a term used to describe the movement of the arm in the plane between the frontal (anterior) and sagittal (lateral) planes. It's essentially a diagonal movement that falls between pure abduction (lifting the arm straight to the side) and flexion (lifting the arm straight forward).

Experimental design

Methods
1. There are some incorrect spellings, e.g., women or women (line 98), and misuse of commas and full stop symbols (e.g., lines 152, 160); please check.
2. Please add more explanation about the normalization method for the strength data used in this review (line 175).
3. Is the starting position important or involved in measuring strength?

Validity of the findings

Results
1. Using the formal form of the Prisma flowchart is better than the present one (Figure 1).
2. Please add more explanation or summarized information about Figure 3 in the texts.
3. Adding the number of articles that use each device to measure shoulder muscle strength, such as shown in (), after the device name is useful (lines 161–164).

Discussion
1. Since the current systematic review focused on the implications of sex-based differences in shoulder strength for ergonomics and task design in the workplace, it would be valuable if the authors added one more paragraph for practical implications. Moreover, providing an example of using current knowledge in terms of ergonomic designs is of interest and helpful.
2. In conclusion, summarizing the findings related to the strength of each motion is helpful for the readers.

---

## Round 0.2 · Minor Revisions

Please consider the remaining reviewer comments.

Reviewer 1 ·

Basic reporting

no comment

Experimental design

no comment

Validity of the findings

no comment

Additional comments

I appreciate the efforts the authors made to revise the manuscript. Overall, they addressed my concerns.

However, I would still suggest being more explicit around the inclusion criteria and/or purpose in some areas. Specifically, I think the focus on single plane movements could be made more clear. This is well explained at lines 177, but I think could be added in other places, i.e. lines 61 and 128.

Reviewer 3 ·

Basic reporting

The revised version contains the most improvements in the background. However, a few points can be improved. For example, using the standard form of the Prisma flowchart provided at http://www.prisma-statement.org/PRISMAStatement/FlowDiagram.aspx is better in terms of providing details for screening the eligible research article than a self-created flowchart. Furthermore, did the authors recognize the influence of limb dominance as one factor related to upper limb strength? This point should be discussed.

Experimental design

-

Validity of the findings

-

---

## Round 0.3 · accepted · Accept

I have assessed the revisions and believe they are sufficient.